# Bioinformatics for Renal and Urinary Proteomics: Call for Aggrandization

**DOI:** 10.3390/ijms21030961

**Published:** 2020-01-31

**Authors:** Piby Paul, Vimala Antonydhason, Judy Gopal, Steve W. Haga, Nazim Hasan, Jae-Wook Oh

**Affiliations:** 1St. Jude Childrens Cancer Research Hospital, 262 Danny Thomas Place, Memphis, TN 38105, USA; piby.paul@stjude.org; 2Department of Microbiology and Immunology, Institute for Biomedicine, Gothenburg University, 413 90 Gothenburg, Sweden; vimalalisha@gmail.com; 3Department of Environmental Health Sciences, Konkuk University, Seoul 143-701, Korea; jejudy777@gmail.com; 4Department of Computer Science and Engineering, National Sun Yat Sen University, Kaohsiung 804, Taiwan; stevewhaga@yahoo.com; 5Department of Chemistry, Faculty of Science, Jazan University, P.O. Box 114, Jazan 45142, Saudi Arabia; nazim7862000@gmail.com; 6Department of Stem Cell and Regenerative Biotechnology, Konkuk University, Seoul 05029, Korea

**Keywords:** omics, renal, urinary proteomics, bioinformatics, databases, tools

## Abstract

The clinical sampling of urine is noninvasive and unrestricted, whereby huge volumes can be easily obtained. This makes urine a valuable resource for the diagnoses of diseases. Urinary and renal proteomics have resulted in considerable progress in kidney-based disease diagnosis through biomarker discovery and treatment. This review summarizes the bioinformatics tools available for this area of proteomics and the milestones reached using these tools in clinical research. The scant research publications and the even more limited bioinformatic tool options available for urinary and renal proteomics are highlighted in this review. The need for more attention and input from bioinformaticians is highlighted, so that progressive achievements and releases can be made. With just a handful of existing tools for renal and urinary proteomic research available, this review identifies a gap worth targeting by protein chemists and bioinformaticians. The probable causes for the lack of enthusiasm in this area are also speculated upon in this review. This is the first review that consolidates the bioinformatics applications specifically for renal and urinary proteomics.

## 1. Introduction

The word ”Proteome” was first used by Marc Wilkins in 1994 at an early Siena proteomic conference [1]. “Proteomics”, according to Anderson and Anderson, is [2] “the use of quantitative protein-level measurement of gene expression to characterize biological processes (e.g., disease processes and drug effects) and decipher the mechanisms of gene expression control”. As this topic has drawn more interest, the field has expanded rapidly, and there has been a surge in the number of published proteomic articles. In 1995, the first publication on proteomics appeared, but as of today, there are approximately 88,300 publications about proteomics, as identified by the Pubmed database. Clinical samples of tissues and biological fluids such as serum, plasma, urine and saliva are all useful sources for diagnostic purposes [3]. 

Urine contains ~2000 proteins [4,5], making it a less complex sample than plasma [6]. The major advantage comes with the fact that the sampling of urine is noninvasive and unrestricted (huge volumes can be obtained easily). Therefore, it is a preferable resource for disease diagnosis. Furthermore, the composition and fragmentation of proteins in urine are relatively more stable, which is an added asset [7]. Urine, being interconnected with blood filtration processes [8,9,10,11,12,13], has been studied for a better understanding of the pathological processes and renal diseases [14]. Serum proteins undergo size-charge dependent filtration at the glomeruli [15], and the reabsorption of serum proteins takes place within the renal tubules [16,17,18,19]. Urinary proteomic studies have led to the identification of candidate biomarkers that are indicative of acute kidney injury, bladder cancer, and diabetic nephropathy (DN) [20,21,22]. Furthermore, because urinary proteins consist of filtered plasma proteins, the urinary proteome is a valuable resource for detecting encephalopathy, heart failure and intestinal ischemia [23,24,25,26,27,28]. The human urine PeptideAtlas database contains 23,739 peptides and 2487 proteins in total [29]. The research curiosity and interest in the area of urinary and renal proteomics is reflected by the dynamic and ever-increasing publications in this area, as shown in Figure 1. 

The predominant analytical techniques that are available for renal and urinary proteomics are mass spectrometry (MS)-based techniques, such as surface enhanced laser desorption/ionization MS (SELDI-MS), liquid chromatography MS (LC–MS), two-dimensional gel electrophoresis MS (2DE-MS), capillary electrophoresis MS (CE–MS) and advanced techniques such as protein microarrays [30]. 

Because of the complexity of proteomic investigations involving various technologies, vast quantities of data are produced. Proteomics has entered a phase of unparalleled growth, as radiated by the large amounts of data outputs. A systematic analysis of the proteomic data has and will continue to offer unprecedented solutions to fundamental questions in biology at the system level. It is in this direction that bioinformatics has offered, with respect to proteomics, effective management, data elaboration, and data integration. Towards this goal, proteomics–bioinformatics integration was introduced. The fundamental role of bioinformatics is thus to reduce the analysis time and to provide validated results. For enabling the smooth processing of data, updated software and algorithms have been developed. These enhance the identification, characterization and quantification of proteins in order to obtain a high-throughput accuracy for acquiring protein information [31]. Furthermore, bioinformatics is useful for guiding functional proteomic studies. Bioinformatics analysis gives vital information on the primary, secondary and tertiary structures of proteins and their alignments and homology, their motifs and domains and their interactions and networks. Several bioinformatics-related analytical tools are freely accessible at http://us.expasy.org/tools. Additional information from bioinformatics analyses facilitates the integration of biomolecular interactions with high throughput data [3]. Ligand-based drug designing in order to modulate metabolic pathways and protein structure, molecular docking and molecular dynamics for structure-based designing for drug discovery have all been enabled through the application of bioinformatics. This has been vital for investigating the impacts on protein folding, stability and function. Bioinformatics brings together the fields of computer science, biology, chemistry, mathematics and engineering for analyzing and interpreting biological information [32]. It is undoubtedly an essential scientific tool. 

The objective of this review is to take a close look at the specific progress made, from the bioinformatics perspective, towards renal and urinary proteomics. The fact that renal and urinary proteomics have not attracted the attention of bioinformaticians is stressed. Given the fact that proteomics has greatly benefitted from a bioinformatic input, it is strange that this crucial area (renal/urinary) of omics is lagging so far behind in terms of the inputs obtainable from bioinformatics. 

## 2. Bioinformatics Repositories for Proteomics

Bioinformatics databases are useful for retrieving biological sequences, structures, compounds and expression profiles, which are the repositories for the computation of biological data. Once such data are retrieved, further analyses can be performed. The proteomic databases that are commonly in use are classified as sequence and structural databases. The most prevalently used protein sequence database is UniProtKB [33] (UniProt Knowledgebase), a storage database for protein sequence and functional information. The cross-references of UniProtKB are connected with other databases, such as UniParc (UniProt Archive), UniRef (UniProt Reference) and UniProt Proteomes. UniParc contains unique cross-references and protein sequences, and UniRef contains clustered sets of sequences from the UniProtKB and specific UniParc records. UniProt Proteomes provides protein sequence IDs and other associated organism details. The web link for this database is www.uniprot.org.

PDB (Protein Data Bank) [34], PDBsum [35] (a pictorial database of 3D structures in PDB) and DisProt (Database of Protein Disorder) [36] are protein structural databases. PDB is the predominant protein three-dimensional (3D) structural database, comprising ~144,000 protein structures. It contains 3D structures of proteins, RNA, DNA, protein–metal ions, protein-drugs and other small molecules (www.rcsb.org). PDBSum provides structural information on the entries in PDB, and is available at http://www.ebi.ac.uk/pdbsum. DisProt provides structural and functional information about intrinsically disordered proteins (IDPs), and is available at www.disprot.org. The 2D gel databases, SWISS-2DPAGE (http://world-2dpage.expasy.org/swiss-2dpage/) [37] and World-2DPAGE (http://world-2dpage.expasy.org/repository/) [38], contain gel-based proteomic information. There are also family and domain databases such as Gene3D (for the structural and functional annotation of protein families; http://gene3d.biochem.ucl.ac.uk/Gene3D/) [39], HAMAP (High-quality Automated and Manual Annotation of Proteins; http://hamap.expasy.org/) [40], InterPro (an integrated resource of protein families, domains and functional sites; http://www.ebi.ac.uk/interpro/) [41], Pfam (a protein families database) [42], PRINTS (a Protein Motif fingerprint database; http://www.bioinf.manchester.ac.uk/dbbrowser/PRINTS/) [43], ProDom (a protein domain families database; http://prodom.prabi.fr/prodom/current/html/home.php) [44], PROSITE (a database of protein domains, families, and functional sites; http://prosite.expasy.org/) [45], TIGRFAMs (The Institute for Genomic Research’s database of protein families; http://www.jcvi.org/cgi-bin/tigrfams/index.cgi) [46], SUPFAM (a superfamily database of structural and functional annotations; http://supfam.org) [47] and SMART (Simple Modular Architecture Research Tool; http://smart.embl.de/) [48]. These aid in locating protein functional regions and their functional information. There are also protein–protein interaction databases that provide molecular interaction details, such as the Database of Interacting Proteins (DIP; http://dip.doe-mbi.ucla.edu/) [49], The Molecular INTeraction database (MINT; http://mint.bio.uniroma2.it/mint/) [50] and STRING (http://string-db.org) [51]. 

The exhaustive list of bioinformatics tools integrated in the Research and Development Sector include the following: FindMod, a potential protein post-translational modification and single amino acid substitution prediction tool; FindPept, a peptide identification tool; Mascot and PepMAPPER, peptide mass fingerprinting tools; ProFound, a protein sequence search tool and ProteinProspector, a tool for analyzing peptide mass data, itself equipped with tools such as MS-Fit, MS-Pattern and MS-Digest. Tools aiding in identification based on the isoelectric point and molecular weight of the protein and its amino acid composition include the following: AACompIdent, AACompSim, TagIdent and MultiIdent. Boinformatic tools that help with protein pattern and profile searches include the following: InterPro Scan, Pfam, PRINTS and MyHits, which reveals the relationships with protein sequences and motifs, as well as scan-based databases such as ScanProsite, HamapScan, MotifScan, Pfam HMM, ProDom, SUPERFAMILY Sequence Search, FingerPRINTScan, PRATT and Eukaryotic Linear Motif (ELM), a resource for functional sites in proteins. Software tools developed for predicting post-translational modifications include the following: ChloroP, LipoP, MITOPROT, PlasMit, Predotar, PTS1, SignalP, DictyOGlyc and NetCGlyc. The elaborate functions of each of these are dealt with elsewhere (ExPASy SIB Bioinformatics Resource Portal - Proteomics Tools.html). 

Proteomic tools for primary structure analysis include ProtParam, Compute pI/Mw and ScanSite pI/Mw. Protein predictor tools include the following: HeliQuest, Radar, REP and Geno3d, which was employed for modelling 3D protein structures. Phyre (upgraded from 3D-PSSM) is a 3D model building tool. Fugue uses sequence-structure homology recognition. HHpred is for protein homology detection and the prediction of a protein structure. LOOPP is for sequence-to-sequence, sequence-to-structure and structure-to-structure alignment, followed by other tools such as PSIpred, MakeMultimer, PQS (Protein Quaternary Structure) and ProtBud, which perform collateral functions. Other important molecular modeling and visualization tools available include Swiss-PdbViewer, SwissDock and SwissParam (ExPASy SIB Bioinformatics Resource Portal - Proteomics Tools.html). Figure 2 gives an overview of the published research in the area of proteomics and bioinformatics, compared with renal and urinary proteomics and bioinformatics.

## 3. Consolidating Available Bioinformatics Tools for Renal and Urinary Proteomics

The bioinformatics resources available for renal and urinary proteomics are summarized in this section. Identification, followed by the characterization of biomarkers/proteins from complex voluminous data, are mandatory for proteomic studies. This is made possible through databases specific for urine and renal proteins. Figure 3 presents the work flow for urinary proteomics, showing the inputs from various bioinformatics resources. Compared with the overflowing abundance of tools available for proteomics as highlighted in Section 2, only a few tools (as shown in Table 1, about 20 databases) specific to the human urine proteome are available. Urine databases include MAPU [52] and Sys-BodyFluid [53]. The Max Planck Institute deployed a proteome database entitled MAPU from sources such as tears, urine, seminal fluid and tissues [52]. MAPU contains information on 1543 proteins. The other relevant proteome database, Sys-BodyFluid, is composed of 11 body fluid proteomes, including urine [53]. This database stores information on the annotation of proteins, as well as on gene ontology, domains, sequences and associated pathways. The human urinary proteomic fingerprint database, UPdb, was released in 2013. There were 200 urine samples tested using SELDI-MS. The database list records 2490 unique peaks/ion species from CE-MS, MALDI and CE-MALDI analyses. To strengthen the human urinary proteome, the “Human Urinary Proteome Database” was constructed using open source technologies, and is available for free at www.urimarker.com/urine [54]. A total of 3,048,648 spectra, 68,151 unique peptides and 6085 proteins are contained in this database. Exhaustive information on the protein name, unique peptide number, accession number, peptide sequence and their sequence coverage are also included. The Human Urinary Proteome Database serves as a good reference repository, including the largest number of urinary proteins.

Other existing urine specific databases are the Urinary Exosome Protein Database, available on http://dir.nhlbi.nih.gov/papers/lkem/exosome/index.html, and the Urinary Protein Biomarker Database, which is available on http://122.70.220.102/biomarker/ [55]. The Mosaiques diagnostics database is a peptidome urinary database comprising more than 13,000 healthy/diseased urine samples obtained by CE-MS [31]. A second part of this database is its biomarker sequence information for hundreds of peptides from 116 proteins [56].

The continued efforts of researchers [57,58,59,60,61] have led to the mapping and deciphering of changes in the urine proteome subsequent to kidney transplantation. The proteomic changes noted in urine during acute rejection (AR) [61], BK virus nephritis (BKVN) [62], chronic allograft nephropathy (CAN) [63] and stable renal allograft (STA) [63] have been serious renal research topics. Through AltAnalyze (www.altanalyze.org), a smaller subset of peptides from proteins that can differentiate between AR, CAN and BKVN transplant injury types [6] has been launched.

In yet other research, a Chinese medicinal herb, *Desmodium styracifolium* (DS), was clinically studied for crystal-induced kidney injuries. A description of this research is instructive in understanding how a combination of methods for network pharmacology and proteomics can be used to explore therapeutic protein targets, in this case, of DS on oxalate crystal-induced kidney injuries [64]. Molecular docking using PharmMapper (http://lilab.ecust.edu.cn/pharmmapper/) helped identify the differential proteins in the three models, so as to acquire differentiated targets. Protein–protein interactions (PPI) were established using STRING. The human structures of these differential proteins were obtained from PDB for docking. Docking was enabled using Discovery Studio 2.5 (http://www.accelrys.com). The active sites of each protein of interest were found from the receptor cavities using the Discovery Studio tool. The docking protocol was performed using the LibDock tool [65]. 

The PharmMapper Server was then employed in this study for the identification of potential targets, by using inverse-docking approaches [66]. The scientific interpretation of the complex relationships between the active components of DS and nephrolithiasis-related protein targets was provided by Cytoscape (http://www.cytoscape.org/). This report clearly highlights the ways various bioinformatics tools come together in conducting a scientific study. 

In recent years, the advancement of bioinformatics tools for the effective analysis of the rapidly increasing proteomics data has been a key area of interest. As part of a large interconnected network, protein and peptide expressions are becoming highly useful for the fundamental understanding of diseases. Van et al. (2017) [67] investigated the biological implications of differentially excreted urinary proteins in patients with diabetic nephrophathy (DN). Artificially constructed PPI networks identified common and stage-specific biological processes in diabetic kidney disease. Data from the Human Protein Atlas were used to study differential protein expressions in kidneys [68]. Data mining techniques have been successfully utilized in diabetes mellitus (DM) [69,70,71,72,73], including clustering, classification and regression models. Thermo raw files were processed using EasierMgf software. Other database searches were enabled using Proteome Discoverer v1.4 (Thermo-Instruments). Based on artificial intelligence and pattern recognition techniques, a therapeutic Performance Mapping System (TPMS; Anaxomics Biotech) [74,75] can integrate the available biological, pharmacological and medical knowledge to simulate human physiology in silico. Databases such as KEGG, BioGRID, IntAct, REACTOME, MINT [51,76,77,78,79] and DrugBank [80,81,82] are valuable assets in this direction. Table 1 consolidates the list of bioinformatics resources available for renal and urinary proteomics.

## 4. Future Perspectives on Bioinformatics Applications: Limitations

Notwithstanding the well-known fact that proteomics is a powerful analytic tool, it still faces innumerable technical limitations. So far, the existing methods for proteomics analysis have only just begun to explore the potential of applying these techniques. Advances in various technologies and the expansion of databases are providing new opportunities to solve proteomic problems, such as for bioinformatics. Urinary proteomics is an ideal target, particularly for human subjects, because it does not require any invasive collection procedures [100]. Normal renal and urinary profiles can be applied to the understanding of renal/urinary diseases. Future directions should focus not only on renal physiology and biomarker detection, but also on new therapies. The integrative analysis of proteomic data and image data has become another hot research area in recent years; the Human Protein Atlas (HPA) aims to map all of the human proteins in cells, tissues and organs using the integration of various omics technologies, including antibody-based imaging. The association analysis of image and protein data has the potential to shed light on the mechanisms of urinary diseases. This is yet another interesting area worth considering with urinary proteomics via bioinformatics, which, again, has not been considered to date. 

With all of this outstanding promise for the future, it was surprising to observe that, to date, there are very few publications reporting proteomic approaches in renal/urinary studies involving bioinformatics tools. Furthermore, only a handful of bioinformatics tools have been released in this area, and most of these were designed and released a decade ago. Our online search disclosed the fact that most research publications reporting urinary/renal proteomic related tools occurred in the period from 2000 to 2015, with only a couple of publications after 2016. It is evident that there is a need for the improvisation and aggrandization of bioinformatics inputs for renal and urinary proteomics. This review hopes to renew researcher interest in this less-worked on area. Input from bioinformaticians and biologists will be needed in order to provide progress for the future perspective of renal and urinary proteomics, but it is an interdisciplinary field, which will also need collaborative coordinated input from software and hardware engineers, molecular biologists, protein chemists, analytical chemists and medical practitioners. There is no doubt that an interdisciplinary approach is key to moving forward the development of new research tools in this area. 

## 5. Conclusions

This review has identified a distinct decline and lack of biocomputational inputs and resources in the area of renal and urinary proteomics. In spite of the fact that urinary samples are some of the easiest to obtain, making them perfect targets for disease detection and prevention, it is surprising that not much active research is available on this area. An upsurge is expected through coordinated input from interdisciplinary researchers. In particular, the surge is expected in the area of software development and tool launches, and testing on renal/urinary clinical samples. This review expects to create awareness and draw more researchers to concentrate on this less explored area.

## Figures and Tables

**Figure 1 ijms-21-00961-f001:**
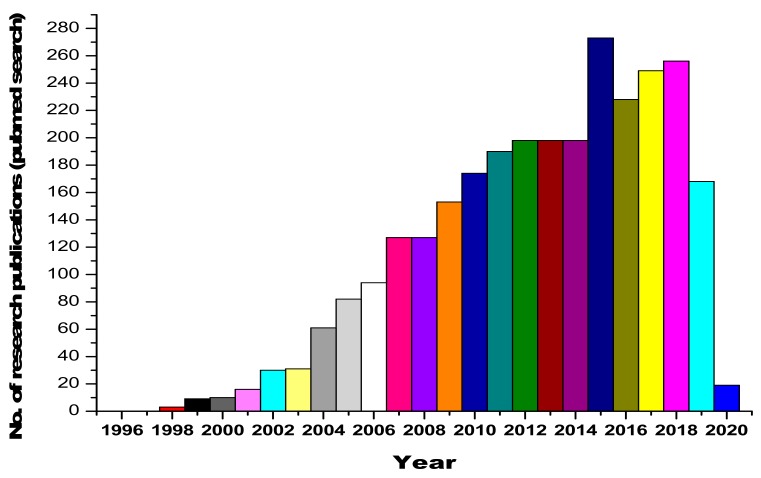
Statistics on research publications related to urinary and renal proteomics, obtained from a Pubmed database search.

**Figure 2 ijms-21-00961-f002:**
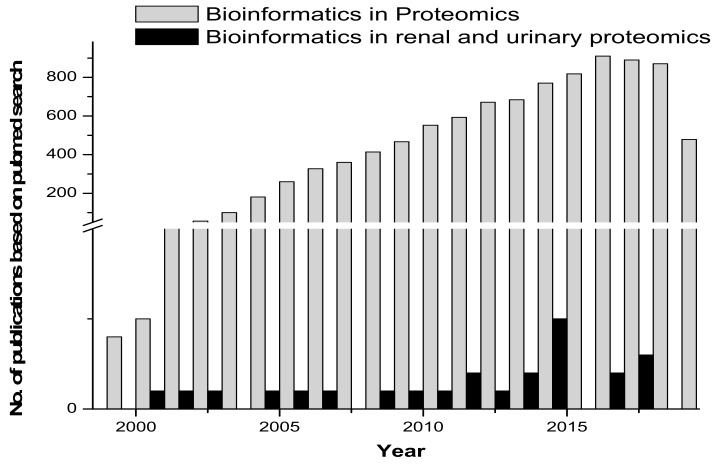
Comparative graph on research published in the area of bioinformatics/biocomputation and proteomics versus bioinformatics/biocomputation and renal and urinary proteomics.

**Figure 3 ijms-21-00961-f003:**
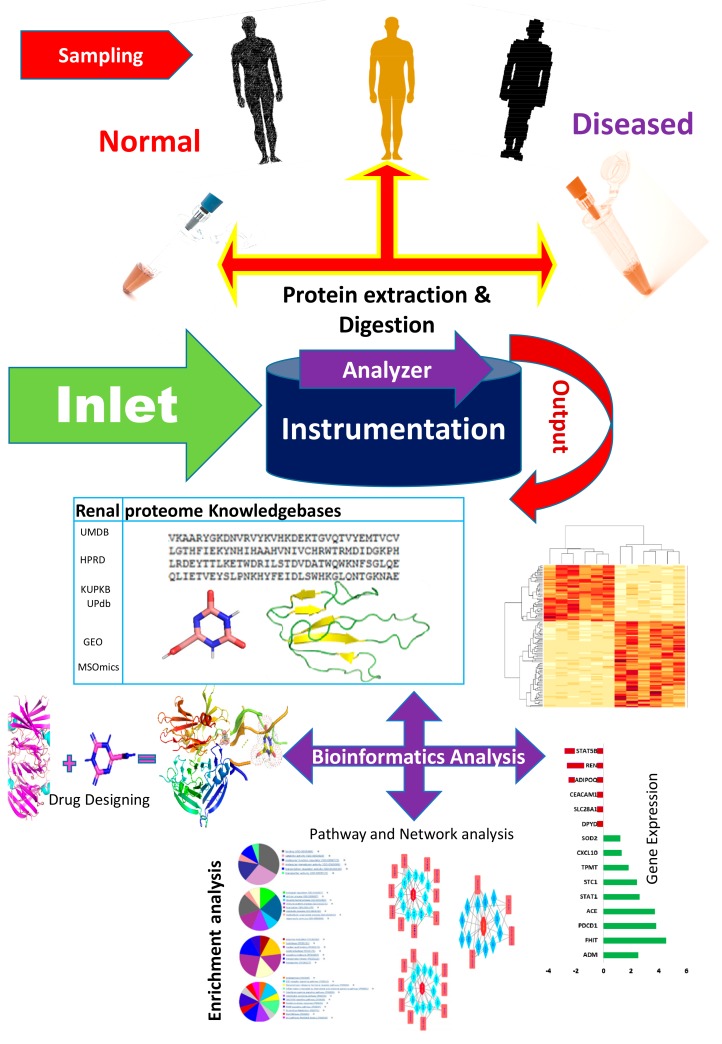
A schematic of the work flow of urinary proteomics research, showing points of interaction where bioinformatics tools are useful. The first step involves the collection of samples from healthy and diseased populations, followed by protein extraction and digestion, prior to analysis using analytical tools. The output data is what is subjected to bioinformatics analysis. UMDB—Urine Metabolome database; HPRD—Human Protein Reference Database; KUPKB—Kidney and Urinary Pathway Knowledge Base; UPdb—Human Urinary Proteomic Fingerprint Database; GEO—Gene Expression Omnibus; MSOmics—The metabolomics service experts.

**Table 1 ijms-21-00961-t001:** Bioinformatics resources for renal and urinary proteomics.

Name	Function	Location	Reference
**Urine Metabolome database (UMDB)**	Metabolites of human urine	http://www.urinemetabolome.ca	Bouatra S et al., PLoS One. 2013 [83]
**Human Urinary Proteomic Fingerprint Database (UPdb)**	Share and exchange primary data derived from SELDI-, MALDI-, material-enhanced laser desorption/ionization (MELDI)-, CE-, LC-, and other TOF-MS analyses in urinary research	http://www.padb.org/	Husi H et al., Int J Proteomics. 2013 [84]
**ProteinProphet^TM^**	Protein identification based on the peptides assigned to the MS/MS spectra	http://proteinprophet.sourceforge.net/	Nesvizhskii AI et al., Anal Chem. 2003 [85]
**PeptideProphet^TM^**	Validates peptide assignments to the MS/MS spectra	http://peptideprophet.sourceforge.net/	Keller A et al., Anal Chem. 2002 [86]
**Urine proteomics for profiling of human disease**	Resource of urinary proteins associated with common and rare human diseases	http://alexkentsis.net/urineproteomics/	Kentsis A et al., Proteomics Clin Appl. 2009 [87]
**Urinary Exosome Protein Database**	Urinary exosomes from healthy human volunteers	https://hpcwebapps.cit.nih.gov/ESBL/Database/Exosome/	Pisitkun T et al., Proc Natl Acad Sci USA. 2004 [9]
**Max-Planck Unified (MAPU) proteome database**	Body fluid (plasma, urine and cerebrospinal fluid) proteomes	http://www.mapuproteome.com/	Zhang Y et al., Nucleic Acids Res. 2007 [52]
**Human Protein Reference Database (HPRD)**	Repository of proteomic information of human proteins	https://www.hprd.org	Marimuthu A et al., J. Proteome. 2011 [88]
**The Kidney and Urinary Pathway Knowledge Base (KUPKB)**	Knowledge related to the kidney and urinary pathways (KUP)	http://www.kupkb.org	Jupp S et al., J Biomed Semantics. 2011 [89]
**Sys-BodyFluid**	Reference database for body fluid proteomics and disease proteomics research	http://www.biosino.org/bodyfluid/	Li SJ et al., Nucleic Acids Res. 2009 [90]
**Gene Expression Omnibus (GEO)**	Gene expression dataset	https://www.ncbi.nlm.nih.gov/geo/	Barrett T, et al., Nucleic Acids Res. 2013 [54]
**Human Kidney and Urine Proteome Project (HKUPP)**	Proteomes of the kidney and urine	http://www.hkupp.org/	Eric W et al., J Proteome Res. 2015 [91]
**Visualization tool for risk factor analysis (VRIFA)**	Computer-aided diagnosis and risk factor analysis	http://dm.postech.ac.kr/vrifa	Cho BH et al., Artif Intell Med. 2008 [92]
**MosaiquesVisu software**	Peak detection, mass deconvolution, 3D data visualizationand generating polypeptide lists	https://www.mhj-tools.com/sps-visu-micro/	Neuhoff et al., Rapid Commun Mass Spectrom. 2004 [93]
**The metabolomics service experts (MSOmics)**	Service provider of metabolomics and for data analysis	http://msomics.com/index.html	Schrimpe A.C. et al., J Am Soc Mass Spectrom. 2016 [94]
**Metabolite set enrichment analysis (MSEA)**	Interprets human metabolite concentration changes in a biologically meaningful context	http://www.msea.ca	Xia J and Wishart D.S. Nucleic Acids Res. 2010 [95]
**Podocyte mRNA Expression Database**	mRNA expression data from FACS-sorted podocytes as analyzed by RNA-sequencing	http://helixweb.nih.gov/ESBL/Database/Podocyte_Transcriptome/index.htm	Kann M et al., J Am Soc Nephrol. 2015 [96]
**MetScape 3.1**	Visualization and interpretation of metabolomic data using Cytoscape	http://metscape.med.umich.edu/	Karnovsky A et al. [97]
**CorrelationCalculator v1.0.1**	Large scale metabolic profiling	http://metscape.med.umich.edu/calculator.html	Basu S et al., Bioinformatics. 2017 [98]
**MetDisease**	MetDisease uses Medical Subject Headings (MeSH) disease terms mapped to PubChem compounds through literature to annotate compound networks.	http://metdisease.ncibi.org/	Duren W et al., Bioinformatics. 2014 [99]

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
