# Peer review of "Bioinformatics for Renal and Urinary Proteomics: Call for Aggrandization"

_ijms, 2020, doi:10.3390/ijms21030961_

Round 1

Reviewer 1 Report

Paul P., et al were reviewed the bioinformatics tools available for urinary proteomics and highlighted decline and lack of biocomputational inputs in the specific area. Current article has interesting collection of work mentioning about several bioinformatics resources and repositories which, is beneficial to researchers in renal disorders and bioinformaticians in urinary proteomics area. Article is well presented including the limitations of the work.

I had no specific suggestions for the authors with exception of the following:

Figure 1: The graph depicts the stats of general term ‘Proteomics’ as mentioned in the text (line:37) but the figure legend says ‘publications related to urinary and renal proteomics’?

Authors are advised to show the graph specific to ‘urinary and renal proteomics’ with updated data.

Figure 2: Authors are suggested to elaborate the figure legend explaining workflow for each stage.

Author Response

Reviewer 1

Paul P., et al were reviewed the bioinformatics tools available for urinary proteomics and highlighted decline and lack of biocomputational inputs in the specific area. Current article has interesting collection of work mentioning about several bioinformatics resources and repositories which, is beneficial to researchers in renal disorders and bioinformaticians in urinary proteomics area. Article is well presented including the limitations of the work.

I had no specific suggestions for the authors with exception of the following:

Ans. We would like to thank the reviewer for the encouraging words. We have now revised the manuscript accodign to your valuable suggestions. Thank you.

Figure 1: The graph depicts the stats of general term ‘Proteomics’ as mentioned in the text (line:37) but the figure legend says ‘publications related to urinary and renal proteomics’?

Ans. Yes rightly pointed out, it was a typo, the description in line 37 was inappropriate, we have now positioned the Figure citation in the appropriate position in the text. Thank you very much.

Authors are advised to show the graph specific to ‘urinary and renal proteomics’ with updated data.

Ans.  Yes, updated. Thank you.

Figure 2: Authors are suggested to elaborate the figure legend explaining workflow for each stage.

Ans. Yes elaborated the figure legend now in Fig 3. Thank you.

Reviewer 2 Report

In this paper, the authors have presented a comprehensive review on the bioinformatics tools for renal and urinary studies. Overall, this is a good review paper, the title, abstract and keywords clearly highlight the aims of the paper. A sufficient literature review has been conducted and the background on the research has been clearly described. The paper is also presented in clear English and an easy way for audiences to follow. However, there exists several issues need to be addressed before publication. 1. Besides the review on urinary proteomics, the author have also provided the comprehensive review on the repositories of other proteomic types in Section 2. The author need to provide more discussions about the associations and differences of these two parts, e.g., which bioinformatics tools introduced in Section 2 are still available for urinary proteomics?. 2. The integrative analysis of proteomic data and image data is another hot research area in recent years, the Human Protein Atlas (HPA) aims to map all the human proteins in cells, tissues and organs using integration of various omics technologies, including antibody-based imaging. The association analysis of image and protein data has the potential to shed light on the mechanism of the urinary disease. The author need to address it in Section 4 as a future research direction.

Author Response

Reviewer 2

In this paper, the authors have presented a comprehensive review on the bioinformatics tools for renal and urinary studies. Overall, this is a good review paper, the title, abstract and keywords clearly highlight the aims of the paper. A sufficient literature review has been conducted and the background on the research has been clearly described. The paper is also presented in clear English and an easy way for audiences to follow. However, there exists several issues need to be addressed before publication.

Ans. Thankyou for your kind words.

Besides the review on urinary proteomics, the author have also provided the comprehensive review on the repositories of other proteomic types in Section 2. The author need to provide more discussions about the associations and differences of these two parts, e.g., which bioinformatics tools introduced in Section 2 are still available for urinary proteomics?.

Ans. We have mentioned the used tools and have highlighted the unused ones in a subtle way. Because except 20 the most part of the tools used for proteomics have still not been applied for urinary proteomics. The review aims at showing this lacunae. Thank you.

The integrative analysis of proteomic data and image data is another hot research area in recent years, the Human Protein Atlas (HPA) aims to map all the human proteins in cells, tissues and organs using integration of various omics technologies, including antibody-based imaging. The association analysis of image and protein data has the potential to shed light on the mechanism of the urinary disease. The author need to address it in Section 4 as a future research direction.

Ans. Yes that a very valuable suggestion, we have added this in Section 4 as per your valuable suggestion. Thank you.

This manuscript is a resubmission of an earlier submission. The following is a list of the peer review reports and author responses from that submission.

Round 1

Reviewer 1 Report

Review the manuscript for English grammar and typographical errors.

Line 70. Not every proteomics analysis leads to significant results. Please modify the statement.

Reviewer 2 Report

This review article is very well written and approachable for both expert and non-bioinformaticist alike. The following points may help to strengthen the manuscript and maximize the response to its call for aggrandization.

It would be helpful to change Figure 1 from a bar graph to an xy-line plot and include other physiological systems as a comparison. Also on Figure 1, the authors may wish to consider truncating the data to only include years which have completed.

Along the same lines, a figure later in the text (perhaps after Figure 2) comparing what has been done using bioinformatics for renal and urinary proteomics against what has been done for other physiological systems would be helpful and might help to inspire readers to start new renal/urinary bioinformatics studies.

Given the current controversies over the use of the term “statistically significant” (e.g., Nature 567, 305-307 (2019); doi: 10.1038/d41586-019-00857-9) the authors may wish to consider the use of an alternative phrase on line 70.

Lines 103-152 provide an exhaustive list of websites that is very useful but makes for difficult reading. Condensing this text and providing a corresponding table (as was done for Table 1) would be helpful.

Reference 53 is a URL, not a journal article. Please be consistent with the placement of URLs either in the body text or in the references.

Line 156 mentions that the use of appropriate statistical methods is mandatory for proteomic studies, but the authors do not provide any clarification of what would be appropriate or inappropriate. The inclusion of some words of wisdom on this topic (with additional references) here would strengthen the manuscript.

On line 159, the authors mention that “few databases specific for [the] human urinary proteome are avail[a]ble” yet there Table 1 appears to list 21 such databases. Clarification here would be helpful.

Figure 2 is of poor resolution. Please increase the resolution to 300 dpi without the use of upsampling.

The acronym DM is used on line 216 without providing a definition. Contextually, it might appear to stand for Data Mining, but that would make the sentence on lines 215-216 nonsensical. Alternatively, it could be a typo for DN, which is defined as Diabetic Nephropathy on line 213. Please clarify.

The authors state on lines 234-235 that it was surprising to find fewer than 50 publications on bioinformatics in renal/urinary proteomics. It would be helpful to readers here to provide context by comparing this number to the number of publications using bioinformatics for one or more other physiological systems. A PubMed search for the term ‘((renal) OR urinary) AND bioinformatics’ returns 6293 articles. Presumably, not all of these are relevant. Nevertheless, it would be helpful for the authors to state in the sentence on lines 234-235 precisely how they determined that there are fewer than 50 publications on this topic.

The conclusions section would be strengthened by adding a specific call to action with clarification of where the authors suggest that attention is most urgently needed in renal/urinary bioinformatics.